# Non-Alcoholic Fatty Liver Disease and Risk of Incident Type 2 Diabetes: Role of Circulating Branched-Chain Amino Acids

**DOI:** 10.3390/nu11030705

**Published:** 2019-03-26

**Authors:** Eline H. van den Berg, Jose L. Flores-Guerrero, Eke G. Gruppen, Martin H. de Borst, Justyna Wolak-Dinsmore, Margery A. Connelly, Stephan J. L. Bakker, Robin P. F. Dullaart

**Affiliations:** 1Department of Endocrinology, University Medical Center Groningen, University of Groningen, 9700 RB Groningen, The Netherlands; e.g.gruppen@umcg.nl (E.G.G.); r.p.f.dullaart@umcg.nl (R.P.F.D.); 2Department of Gastroenterology and Hepatology, University Medical Center Groningen, University of Groningen, 9700 RB Groningen, The Netherlands; 3Department of Nephrology, University Medical Center Groningen, University of Groningen, 9700 RB Groningen, The Netherlands; j.l.flores.guerrero@umcg.nl (J.L.F.-G.); m.h.de.borst@umcg.nl (M.H.d.B.); s.j.l.bakker@umcg.nl (S.J.L.B.); 4Laboratory Corporation of America Holdings (LabCorp), Morrisville, NC 27560, USA; wolakdj@labcorp.com (J.W.-D.); connem5@labcorp.com (M.A.C.)

**Keywords:** branched-chain amino acids, type 2 diabetes, non-alcoholic fatty liver disease, fatty liver index, insulin resistance

## Abstract

Non-alcoholic fatty liver disease (NAFLD) is likely to be associated with elevated plasma branched-chain amino acids (BCAAs) and may precede the development of type 2 diabetes (T2D). We hypothesized that BCAAs may be involved in the pathogenesis of T2D attributable to NAFLD and determined the extent to which plasma BCAAs influence T2D development in NAFLD. We evaluated cross-sectional associations of NAFLD with fasting plasma BCAAs (nuclear magnetic resonance spectroscopy), and prospectively determined the extent to which the influence of NAFLD on incident T2D is attributable to BCAA elevations. In the current study, 5791 Prevention of REnal and Vascular ENd-stage Disease (PREVEND) cohort participants without T2D at baseline were included. Elevated fatty liver index (FLI) ≥60, an algorithm based on triglycerides, gamma-glutamyltransferase, body mass index (BMI) and waist circumference, was used as proxy of NAFLD. Elevated FLI ≥ 60 was present in 1671 (28.9%) participants. Cross-sectionally, BCAAs were positively associated with FLI ≥ 60 (β = 0.208, *p* < 0.001). During a median follow-up of 7.3 years, 276 participants developed T2D, of which 194 (70.2%) had an FLI ≥ 60 (log-rank test, *p* < 0.001). Cox regression analyses revealed that both FLI ≥60 (hazard ratio (HR) 3.46, 95% CI 2.45–4.87, *p* < 0.001) and higher BCAA levels (HR 1.19, 95% CI 1.03–1.37, *p* = 0.01) were positively associated with incident T2D. Mediation analysis showed that the association of FLI with incident T2D was in part attributable to elevated BCAAs (proportion mediated 19.6%). In conclusion, both elevated FLI and elevated plasma BCAA levels are associated with risk of incident T2D. The association of NAFLD with T2D development seems partly mediated by elevated BCAAs.

## 1. Introduction

Non-alcoholic fatty liver disease (NAFLD) is emerging as the most common cause of chronic liver disease in the Western world [1]. NAFLD is characterized by hepatic steatosis in the absence of alcohol abuse and its spectrum ranges from simple steatosis to non-alcoholic steatohepatitis (NASH), fibrosis and cirrhosis [1]. NAFLD and insulin resistance are closely related [2]. NAFLD frequently coincides with the metabolic syndrome (MetS) and type 2 diabetes (T2D). While MetS and T2D may precede NAFLD [2,3], two recent meta-analyses have demonstrated that NAFLD, irrespective of whether it is diagnosed by elevated liver enzymes, by radiological abnormalities or by histological abnormalities, may in fact precede T2D [4,5]. Moreover, an elevated fatty liver index (FLI), as proxy of NAFLD and assessed by an algorithm based on obesity measures, plasma triglycerides and gamma-glutamyltransferase (GGT), has been shown to predict T2D in two independent European populations [6,7]. Taken together, these findings [2,4,5,6,7] point to an intricate relationship between NAFLD, MetS and T2D, which are indeed considered as manifestations of a common cardio-metabolic multisystem disorder [8,9,10,11].

Branched-chain amino acids (BCAAs) are amino acids with non-linear aliphatic side-chains, and include the essential amino acids leucine, valine and isoleucine [12]. Although not fully understood, BCAAs may contribute to the development of obesity-associated insulin resistance [13,14]. Moreover, in the clinical setting, disturbances in BCAA metabolism have been described in insulin-resistant states, including MetS and T2D [15,16,17,18,19,20,21]. Conversely, a decrease in BCAA levels may result in an improvement in glucose metabolism [22]. Additionally, high plasma BCAA levels have been shown to be associated with an increased risk of T2D development [15].

Fasting plasma BCAAs may be elevated in (obesity-associated) NAFLD [13,23,24,25,26], and may coincide with abnormalities in BCAA catabolic enzymes in liver and adipose tissue [27]. Such abnormalities in (hepatic) BCAA metabolism conceivably affect carbon substrate oxidation that, together with impairment of antioxidant defence, may contribute to the generation of reactive oxygen species [13,19]. In turn, mitochondrial dysfunction in NAFLD, in particular in the context of hepatic inflammation, is likely to convey impaired BCAA metabolism conceivably resulting in plasma BCAA elevations [28,29,30]. Thus, it seems plausible to hypothesize that NAFLD and impaired hepatic BCAA metabolism jointly impact on deteriorating glucose tolerance.

Despite continued clinical interest concerning the impact of NAFLD and increased circulating BCAAs on the development of T2D, no large-scale population-based studies have yet reported on the relative contributions of NAFLD and plasma BCAAs on incident T2D in initially diabetes-naïve NAFLD subjects. Therefore, we initiated the present study to determine the extent to which BCAAs influence T2D development in the context of NAFLD. To this end, we carried out cross-sectional and prospective analyses among 5791 subjects participating in the Prevention of REnal and Vascular ENd-stage Disease (PREVEND) cohort, comprising a large and well-characterized population from the North of the Netherlands.

## 2. Materials and Methods

### 2.1. Study Population

The study was performed among participants of the Prevention of REnal and Vascular ENd-stage Disease (PREVEND) cohort study [31,32]. The PREVEND study was approved by the Medical Ethics Committee of the University Medical Center Groningen and performed in accordance with the Declaration of Helsinki guidelines [31,32]. All participants gave written informed consent. PREVEND is a large prospective general population-based study that was initiated to investigate cardiovascular and renal disease with a focus on albuminuria. All inhabitants (28–75 years old) of Groningen, The Netherlands were sent a questionnaire on demographics and cardiovascular morbidity and were asked to supply an early morning urine specimen. Pregnant women, type 1 diabetic subjects and T2D subjects using insulin were not allowed to participate. All participants with a urinary albumin concentration of ≥10 mg/L were invited to our clinic together with randomly selected subjects with a urinary albumin concentration of <10 mg/L. The initial study population of the PREVEND study was comprised of 8592 subjects who completed the total study screening program.

For the present study, we conducted a post-hoc analysis, using data of participants who completed the second screening round (*n* = 6893). We excluded all subjects with missing values of BCAA concentrations, pre-existing T2D, non-fasting subjects, and subjects in which the clinical and biochemical variables required to calculate the fatty liver index (FLI), a proxy of NAFLD, were not available, leaving a study population of 5791 participants with complete information for analysis.

### 2.2. Measurements and Definitions

During two outpatient visits, baseline data were collected on demographics, lifestyle factors, anthropometric measurements, medical history, parental history of T2D and medication use. Information on medication use was combined with information from a pharmacy-dispensing registry, which had complete information on the drug usage of >95% of subjects in the PREVEND study. Height and weight were measured in standing position without shoes and heavy outer garments. Body mass index (BMI) was calculated as weight (kg) divided by height squared (meter). Waist circumference was measured as the smallest girth between the rib cage and iliac crest. The waist/hip ratio was determined as the waist circumference divided by the largest girth between waist and thigh [31]. Blood pressure was measured using an automatic device, where the last two recordings of the second outpatient visit were averaged for analysis. Alcohol consumption was recorded with one alcoholic drink being assumed to contain 10 g of alcohol. Smoking was categorized into current and never/former smokers. Past cardiovascular history included hospitalization for myocardial ischemia, obstructive coronary artery disease or revascularization procedures. Homeostasis Model Assessment (HOMA)-IR was calculated as follows:
Fasting plasma insulin (mU/L) × fasting plasma glucose (mmol/L)/22.5.(1)

HOMA-β was calculated as follows:
20 × Fasting plasma insulin (mU/L)/(fasting plasma glucose (mmol/L) − 3.5,(2)
where HOMA-β represents the relative β-cell function (expressed as a percentage).

Urinary albumin excretion (UAE) was measured as described in two 24 h urine collections and the results were averaged for analysis [31]. Estimated glomerular filtration rate (eGFR) was calculated applying the combined creatinine cystatin C-based Chronic Kidney Disease Epidemiology Collaboration equation.

For the diagnosis of NAFLD, the algorithm of the FLI was used [33]. The FLI was calculated according to the following formula:
[e (0.953 × loge (triglycerides + 0.139 × BMI + 0.718 × loge (GGT) + 0.053 × waist circumference − 15.745)]/[1 + e (0.953 × loge (triglycerides) + 0.139 × BMI + 0.718 × loge (GGT) + 0.053 × waist circumference − 15.745)] × 100.(3)

The optimal cut-off value for the FLI is documented to be 60 with an accuracy of 84%, a sensitivity of 61% and a specificity of 86% for detecting NAFLD as determined by ultrasonography [33]. FLI ≥60 was therefore used as proxy of NAFLD. The FLI is currently considered as one of the best-validated steatosis scores for larger scale screening studies [34]. Alternatively, we used the hepatic steatosis index (HSI) [35]. The HSI is defined as follows:
HSI = 8 × ALT/AST ratio + BMI (+2, if diabetes; +2, if female),(4)
where ALT is alanine aminotransferase and AST is aspartate aminotransferase. The cut-off value of the HSI for detecting NAFLD is 36 [35].

The MetS was defined according to the revised National Cholesterol Education Program Adult Treatment Panel (NCEP-ATP) III criteria [36]. Participants were categorized with MetS when at least three out of the following five criteria were present: waist circumference >102 cm for men and >88 cm for women; plasma triglycerides ≥1.7 mmol/L; high-density lipoprotein (HDL) cholesterol <1.0 mmol/L for men and <1.3 mmol/L for women; hypertension (blood pressure ≥130/85 mm Hg or the use of antihypertensive medication); hyperglycaemia (fasting glucose ≥5.6 mmol/L or the use of glucose-lowering drugs).

### 2.3. Laboratory Methods

Venous blood samples were drawn after an overnight fast while participants rested for 15 min. Heparinized plasma and serum samples were obtained by centrifugation at 1400× *g* for 15 min at 4 °C. Plasma and serum samples were stored at −80 °C until analysis. EDTA plasma valine, leucine and isoleucine concentrations were measured using a Vantera Clinical Analyzer (LabCorp., Morrisville, NC, USA), a fully automated, high-throughput, 400 MHz proton (1H) nuclear magnetic resonance (NMR) spectroscopy platform. Plasma samples were prepared on board the instrument, and automatically delivered to the flow probe in the NMR spectrometer’s magnetic field. The validation of the use of NMR for quantification of BCAAs has been previously described [20,21]. Data acquisition on the Vantera and the spectra data processing have been reported in greater detail elsewhere [37]. Fasting plasma glucose was measured by dry chemistry (Eastman Kodak, Rochester, NY, USA) directly after blood collection. Insulin was measured with an immunoturbidometric assay (Diazyme Laboratories, Poway, CA, USA). Plasma total cholesterol, triglycerides and HDL cholesterol were measured as previously described [31,32]. Non-HDL cholesterol was calculated as the difference between total cholesterol and HDL cholesterol. Low-density lipoprotein (LDL) cholesterol was calculated by the Friedewald formula if triglycerides were <4.5 mmol/L. Serum ALT and AST were measured using the standardized kinetic method with pyridoxal phosphate activation (Roche Modular P, Roche Diagnostics, Mannheim, Germany). Serum GGT was assayed by an enzymatic colorimetric method (Roche Modular P, Roche Diagnostics, Mannheim, Germany). Standardization of ALT, AST and GGT was performed according to the International Federation of Clinical Chemistry guidelines [38,39,40]. High sensitivity C-reactive protein (hsCRP) was assayed by nephelometry (Dade Behring Diagnostic, Marburg, Germany). Serum creatinine was measured by an enzymatic method on a Roche Modular analyzer (Roche Diagnostics, Mannheim, Germany). Serum cystatin C was measured using Gentian Cystatin C Immunoassay (Gentian AS, Moss, Norway) reagents on a modular analyzer (Roche Diagnostics). Urinary albumin was measured by nephelometry (Dade Behring Diagnostic, Marburg, Germany).

### 2.4. T2D Development

Participants were followed from baseline outpatient visit until end of the follow-up period. Incident T2D was established if one or more of the four criteria were met during follow-up:

(1) Blood glucose ≥7.0 mmol/L (126 mg/dL); (2) Random sample plasma glucose ≥11.1 mmol/L (200 mg/dL); (3) Self-report of a physician diagnosis; (4) Initiation of glucose-lowering medication according to the central pharmacy registry follow-up data, which was completed on 1 January 2011.

### 2.5. Statistical Analysis

IBM SPSS software (version 23.0, IBM Corp., Armonk, NY, USA) was used for cross-sectional data analysis. R version 3.4.2 (Boston, MA, USA) and STATA version 13.1 (StataCorp LP, College Station, TX, USA) were used for prospective analyses. Cross-sectional data are expressed as mean ± standard deviation (SD), median with interquartile range (IQR) or as numbers (percentages). HOMA-IR and HOMA-β were log_e_ transformed for multivariable regression analyses. Normality of distribution was assessed and checked for skewness. Between-group differences in variables were determined by unpaired *t*-tests for normally distributed and log_e_ transformed variables, by Mann–Whitney U tests for non-normally distributed variables or by chi-squared tests for categorical variables where appropriate. Multivariable linear regression analyses were carried out to disclose the independent associations of BCAA levels with an elevated FLI or HSI while taking into account clinical covariates and laboratory parameters. Results from multivariable linear regression analyses are presented as standardized regression coefficients.

For the prospective analyses, we plotted cumulative Kaplan–Meier curves for T2D development during follow-up according to quartiles of plasma total BCAAs. Time-to-event Cox proportional hazards models were used to assess the hazard ratio (HR) with 95% confidence intervals (CI) of incident T2D.

A mediation analysis was performed to disclose the extent to which the plasma total BCAA concentration was a possible mediator between an elevated FLI and HSI, as proxies of NAFLD, and incident T2D, following the procedures as advocated by Preacher and Hayes [41,42]. The significance of the mediation effect was tested by computing bias-corrected bootstrap CIs with 2000 repetitions. Finally, the magnitude of mediation was calculated by dividing the coefficient of the indirect effect by the total effect. Significance of mediation was proved with *p* < 0.05 if zero was not between the lower and upper bound of the 95% CI of the indirect effect. Interaction terms were considered to be statistically significant at two-sided *p*-values <0.10 [43]. Otherwise, two-sided *p*-values <0.05 were considered significant.

## 3. Results

### 3.1. Clinical and Laboratory Characteristics of the Study Population

The study population consisted of 5791 participants free of T2D at baseline. There were 1671 participants (28.9%) with an FLI ≥60. Table 1 shows the clinical characteristics and laboratory data of the study population according to FLI categorization. Subjects with an FLI ≥60 were older, more likely to be men (67.8% vs. 32.2%) and more likely to be classified with MetS, history of cardiovascular disease and parental history of T2D. Consequently, subjects with an FLI ≥60 used antihypertensive medication and lipid-lowering drugs more frequently. Alcohol consumption ≥10 g/day was recorded in subjects with an elevated FLI more frequently, but cigarette smoking was not significantly different. BMI, waist circumference, waist/hip ratio, systolic and diastolic blood pressure, glucose, insulin, HOMA-IR, HOMA-β, hsCRP, transaminases, ALP, GGT, UAE, total cholesterol, non-HDL cholesterol, LDL cholesterol and triglycerides were higher in subjects with FLI ≥60, whereas eGFR and HDL cholesterol were lower in subjects with an elevated FLI (Table 1). Plasma total BCAAs as well as valine, leucine and isoleucine concentrations were all higher in subjects with an FLI ≥60 (Table 1).

### 3.2. Cross-Sectional Associations of BCAA with an Elevated FLI and HSI

Multivariable linear regression analyses were subsequently performed in order to establish the extent to which plasma BCAA concentrations were independently associated with an elevated FLI (Table 2). In an age- and sex-adjusted analysis, a positive association of plasma BCAA concentrations with an elevated FLI was found (Table 2, Model 1, β = 0.326, *p* < 0.001). This positive association of plasma BCAA concentrations with an elevated FLI was also demonstrated after further adjustment for family history of T2D (Table 2, Model 2, β = 0.324, *p* < 0.001), alcohol intake and current smoking (Table 2, Model 3, β = 0.323, *p* < 0.001), eGFR, UAE, use of antihypertensive medication and lipid-lowering drugs (Table 2, Model 4, β = 0.318, *p* < 0.001) and, finally, after additional adjustment for HOMA-IR and HOMA-β (Table 2, Model 5, β = 0.208, *p* < 0.001). In alternative analyses with an elevated HSI instead of an elevated FLI, similar independent positive associations of plasma BCAA concentrations with an elevated HSI were found (Appendix A, all models *p* < 0.001).

### 3.3. Prospective Analyses of FLI and BCAA with Incident T2D

During a median follow-up of 7.3 years (IQR 5.7–7.7), a total of 276 participants developed T2D, of which 194 (70.2%) subjects had an elevated FLI (Table 3). Figure 1 shows the Kaplan–Meier curves for incident T2D survival according to non-elevated FLI (<60) and elevated FLI (≥60), which shows a significantly increased risk of incident T2D in subjects with an elevated FLI (log-rank test *p* < 0.001). Comparing associations of non-elevated and elevated FLI with incident T2D in Cox regression analyses showed an HR of 5.84 (95% CI 4.46–7.66, *p* < 0.001) when adjusted for age and sex (Table 3). This positive association of elevated FLI with risk of T2D incidence remained present after further adjustment for family history of T2D, HR 5.72 (95% CI 4.37–7.50, *p* < 0.001); alcohol intake and smoking, HR 5.64 (95% CI 4.31–7.39, *p* < 0.001); eGFR, UAE, antihypertensive medication and lipid-lowering drugs, HR 5.09 (95% CI 3.77–6.88, *p* < 0.001); HOMA-IR and HOMA-β, HR 3.84 (95% CI 2.76–5.36, *p* < 0.001); and, finally, for BCAAs, HR 3.46 (95% CI 2.45–4.87, *p* < 0.001) (Table 3). Similar Cox regression analyses were performed to address the relationship between BCAA (per 1 SD increment) and incident T2D (Table 4), which showed a similar pattern with T2D risk. However, in the fully adjusted model 6, additional adjustment for an elevated FLI attenuated the hazard ratio to 1.19 (95% CI 1.03–1.37, *p* = 0.01) (Table 4). There was no significant interaction of elevated FLI with age (*p* = 0.11) and there was a marginally significant interaction with sex (*p* = 0.07). In Cox regression analyses in which an elevated HSI was used instead, an elevated HSI was similarly associated with incident T2D, which remained associated with incident T2D independent of adjustment for plasma total BCAAs, whereas in analyses with plasma total BCAAs as determinant of incident T2D, the association of plasma total BCAAs with incident T2D was reduced after adjustment for an elevated HSI (Appendix A). There was no significant interaction of sex with an elevated HSI on incident T2D (*p* = 0.36 for sex interaction) and there was a marginally significant interaction of elevated HSI with age (*p* = 0.07).

Additionally, mediation analyses were performed to disclose the extent to which the plasma total BCAA concentration was a possible mediator between elevated FLI or HSI and incident T2D. BCAAs appeared to be a mediator in the association of FLI with incident T2D (Figure 2). The indirect pathway was significant (B = 0.040, 95% CI 0.027–0.054, *p*-indirect < 0.001) and the magnitude of mediation was 19.6% (Table 5). In alternative analyses, BCAAs appeared to be a mediator in the association of HSI with incident T2D (Figure 3). The indirect pathway was significant (B = 0.033, 95% CI 0.026–0.041, *p*-indirect < 0.001) and the magnitude of mediation was 22.6% (Table 6).

## 4. Discussion

In this large-scale study among a predominantly Caucasian population, who were free of T2D at the baseline evaluation, we have first cross-sectionally demonstrated that fasting total plasma BCAA levels are elevated in subjects with NAFLD and that this relationship of BCAAs with NAFLD is independent of a considerable number of clinical and laboratory covariates, including measures of insulin resistance and β-cell function. Second, we have longitudinally documented that an elevated FLI predicts the development of T2D, an association that was in part attributable to higher plasma total BCAAs as inferred from mediation analyses. Taken together, the current results not only indicate that there is an intricate relationship between NAFLD and elevated plasma BCAA levels, but also suggest that NAFLD and elevated circulating BCAAs could act together on T2D development. In this study, we have used the FLI [33] as proxy of NAFLD, which is in line with recommendations of international guidelines [34], advocating the use of biomarker-derived algorithms in order to categorize subjects with probable NAFLD in large-scale studies. Alternative analyses using the HSI as proxy of NAFLD [35] confirmed the findings, supporting the validity of our results.

The cross-sectional part of our study demonstrated that plasma BCAA elevations were independently associated with suspected NAFLD, irrespective of whether an elevated FLI or an elevated HSI was used as a proxy of NAFLD. In comparison, recent studies have also shown that plasma BCAAs are elevated in the context of (obesity-associated) NAFLD, and that BCAAs may aggravate mitochondrial dysfunction in NAFLD [13,19]. Obesity, MetS and T2D are all related to plasma BCAA elevations [24], and elevated BCAA levels in NAFLD are likely linked to increased insulin resistance and protein catabolism [13,24]. Accumulating evidence points to a key role of mitochondrial dysfunction in the pathophysiology and progression of NAFLD, possibly caused by increased mitochondrial β-oxidation of fatty acids in insulin-resistant states [28,29] and by impaired adaptation of hepatic mitochondrial function in NAFLD [30]. A dysfunctional hepatic tricarboxylic acid (TCA) cycle is regarded as a central feature of hepatic insulin resistance. BCAAs are essential to mediate the transport of carbon substrates for oxidation through the mitochondrial TCA cycle, and an impaired upregulation of BCAA-mediated TCA is thought to be a significant contributor of mitochondrial dysfunction in NAFLD [19]. Moreover, the hepatic accumulation of BCAAs is mainly regulated by the activities of transporters including the SLC43A1 (LAT3), which are responsible for controlling the efflux of BCAAs from the liver to the circulation [44]. In addition to the accumulation of BCAAs in hepatocytes, hepatic BCAA-degrading enzymes are downregulated during worsening of NAFLD [45]. Taken together, it seems plausible that hepatic fat accumulation and altered BCAA metabolism have a negative impact on each other in worsening of NAFLD and further deteriorating BCAA metabolism and their accumulation in the circulation. Notably, the association of plasma BCAA elevations with suspected NAFLD, as cross-sectional documented in the current study, was found to be independent of HOMA-IR and β-cell function. We did not adjust for BMI or waist circumference in the multivariable analysis because obesity indices comprise part of the FLI and HSI algorithms. Such an association of plasma BCAA elevations with suspected NAFLD would suggest that maladaptation in fatty liver may contribute to mitochondrial dysfunction [13], resulting in plasma BCAA elevations.

Our findings regarding the longitudinal association of an elevated FLI with incident T2D was anticipated. Of note, a similar association was found in alternative analysis using an elevated HSI instead. Two recent smaller-scaled studies among subjects of European ancestry, namely, the European Prospective Investigation into Cancer and Nutrition-Potsdam study and a cohort of Spanish adults with pre-diabetes, have demonstrated that an elevated FLI is associated with increased risk of T2D development [6,7]. We also anticipated that plasma BCAA elevations predicted T2D risk [15]. However, to the best of our knowledge, our study shows for the first time that the association of suspected NAFLD with incident T2D appears to be in part attributable to plasma BCAA elevations. In the current study, mediation analysis suggested that 19.6% of the association of an elevated FLI and, alternatively, 22.6% of the association of an elevated HSI with incident T2D was mediated by plasma BCAA concentrations. Hepatic fat accumulation has a deleterious role in T2D development [4,46], and recent evidence shows accumulation of BCAAs in the liver to take place in the context of progressive steatohepatitis [47]. Although our study supports the hypothesis that higher BCAA levels promote the development of T2D, it is clear that further research is needed regarding the mechanisms whereby altered BCAA metabolism and NAFLD may act together to deteriorate glucose tolerance, an effect which seems, at least in part, independent of insulin resistance as demonstrated here with respect to both elevated FLI- and BCAA-associations with incident T2D.

Our study has several strengths. To the best of our knowledge, this is the first study reporting on the joint contributions of NAFLD and higher total plasma BCAAs on the development of T2D. Furthermore, considering a sample size of over 5500 individuals, this is the largest study to date reporting on the association of both suspected NAFLD and plasma BCAA levels with T2D development, which enabled us to carry out sufficiently powered multivariable adjusted analyses. Furthermore, we also used a robust methodology, adjusting our results for relevant variables such as HOMA-IR to investigate the association between FLI and incident T2D. Several other methodological aspects and limitations also need to be addressed. First, the PREVEND cohort study mainly comprises individuals of European ancestry, which could limit extrapolation of our findings to other ethnicities. Second, the FLI is not an absolute measure of hepatic fat accumulation and thus some over- and underestimation of NAFLD could have occurred. However, the FLI is considered to have sufficient accuracy for NAFLD assessment and has been validated against magnetic resonance spectroscopy with moderate diagnostic accuracy for NAFLD [48]. Indeed, the use of the FLI is in line with international guidelines to apply biomarker scores in order to characterize NAFLD in larger-sized cohorts [33,34]. Additionally, liver biopsy, with well-known limitations with respect to invasiveness and sampling variability, or liver ultrasound were not feasible in the PREVEND cohort study, which recruited individuals from the general population. Moreover, the positive associations of plasma BCAA levels with suspected NAFLD and prospective associations with incident T2D were confirmed using the HSI as an alternative algorithm for NAFLD categorization [35]. Third, we did not have measurements of insulin and BCAA levels beyond baseline assessment, which limited us to evaluate the evolution of insulin resistance and regression dilution of BCAAs could not be excluded. Therefore, underestimation of the BCAA–incident T2D associations could have occurred. Fourth, the proportion of subjects using alcohol in excess of 30 g per day in the PREVEND is low (5.2%) [49], and we adjusted for alcohol consumption in all analyses. Finally, people with micro-albuminuria preferentially participated in the PREVEND cohort and therefore multivariable and prospective analyses were adjusted for eGFR and UAE, showing positive and independent associations of plasma BCAA levels in NAFLD and incident T2D development.

## 5. Conclusions

This large-scale population study cross-sectionally demonstrated that elevated plasma BCAA levels are positively associated with an elevated FLI, as a proxy of NAFLD. Furthermore, it was longitudinally shown that the association of NAFLD with T2D development is likely to be attributable in part to plasma BCAA elevations, which likely reflects abnormalities in BCAA metabolism.

## Figures and Tables

**Figure 1 nutrients-11-00705-f001:**
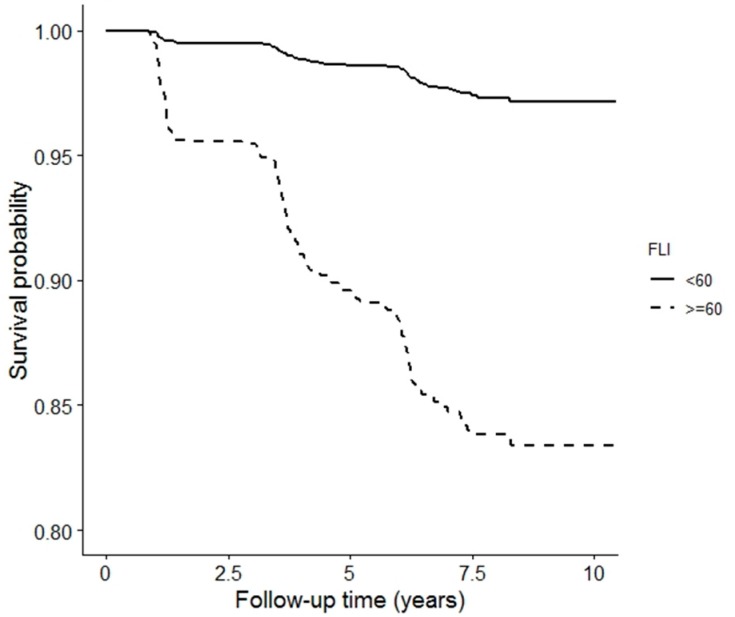
Kaplan–Meier curves for incident type 2 diabetes survival according to FLI score. Log-rank test (*p* < 0.001). FLI, fatty liver index.

**Figure 2 nutrients-11-00705-f002:**
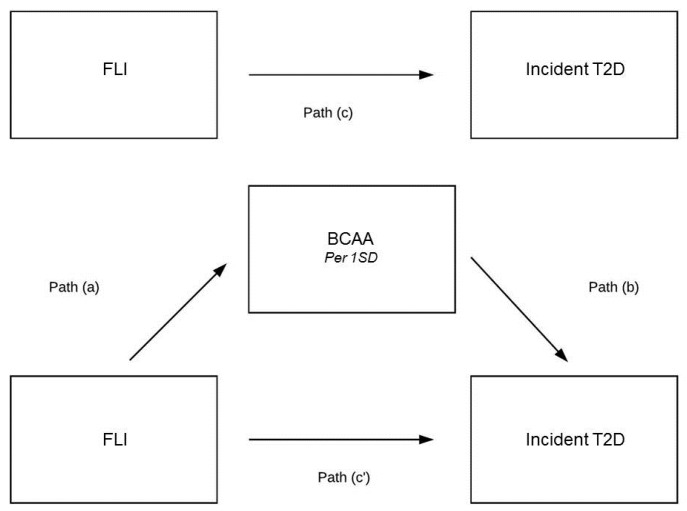
Mediation analysis on the association of FLI with incident type 2 diabetes. (a)–(c) are the regression coefficients between variables. The indirect effect is calculated as a × b. Total effect (c) is a × b + c’. Magnitude of mediation is calculated as indirect effect divided by total effect.

**Figure 3 nutrients-11-00705-f003:**
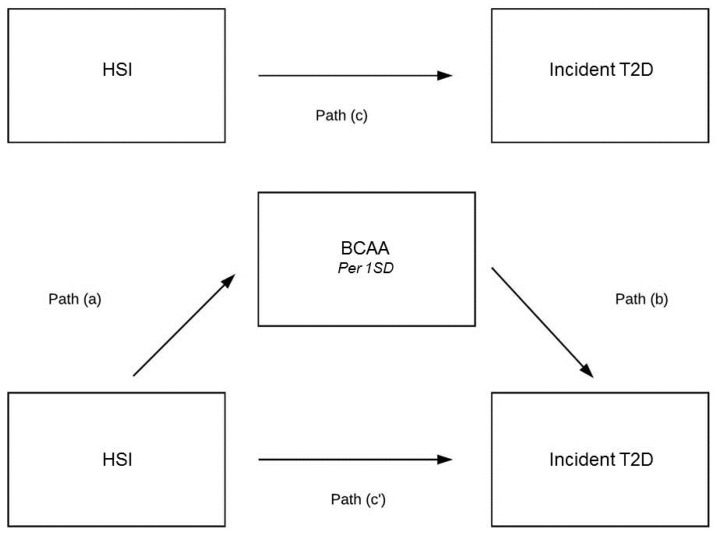
Mediation analysis on the association of HSI with incident type 2 diabetes. (a)–(c) are the regression coefficients between variables. The indirect effect is calculated as a × b. Total effect (c) is a × b + c’. Magnitude of mediation is calculated as indirect effect divided by total effect.

**Table 1 nutrients-11-00705-t001:** Clinical and laboratory characteristics including plasma branched-chain amino acids in 4120 subjects with a fatty liver index (FLI) < 60 and 1671 subjects with an FLI ≥60.

	FLI < 60, *n* = 4120 (71.1%)	FLI ≥ 60, *n* = 1671 (28.9%)	*p*-Value
Age (years), median (IQR)	49.6 (41.9–59.3)	56.0 (47.7–65.6)	<0.001
Sex (men/women), *n* (%)	1707 (41.4)/2413 (58.6)	1133 (67.8)/538 (32.2)	<0.001
MetS, *n* (%)	296 (7.2)	946 (56.6)	<0.001
History of cardiovascular disease, *n* (%)	169 (4.1)	155 (9.3)	<0.001
Parental history of T2D, *n* (%)	577 (14.0)	266 (15.9)	0.061
Current smokers, *n* (%)	1167 (28.3)	457 (27.3)	0.454
Alcohol ≥10 g/day, *n* (%)	137 (3.4)	109 (6.6)	<0.001
Antihypertensive medication, *n* (%)	560 (13.6)	556 (33.3)	<0.001
Lipid-lowering drugs, *n* (%)	249 (6.0)	234 (14.0)	<0.001
Systolic blood pressure (mm Hg), mean ± SD	121 ± 17	134 ± 18	<0.001
Diastolic blood pressure (mm Hg), mean ± SD	71 ± 8	77 ± 9	<0.001
BMI (kg/m^2^), mean ± SD	24.7 ± 2.8	30.8 ± 4.0	<0.001
Waist circumference, mean ± SD	85.8 ± 9.1	104.8 ± 8.9	<0.001
Waist/hip ratio, mean ± SD	0.87 ± 0.07	0.96 ± 0.07	<0.001
Glucose (mmol/L), mean ± SD	4.71 ± 0.58	5.10 ± 0.67	<0.001
Insulin (mU/L), median (IQR)	6.80 (5.1–9.2)	12.50 (9.2–18.1)	<0.001
HOMA-IR (mU mmol/L^2^/22.5), median (IQR)	1.42 (1.04–1.98)	2.86 (2.00–4.17)	<0.001
HOMA-β (%), median (IQR)	25.55 (18.61–35.31)	46.92 (33.29–66.68)	<0.001
hsCRP (mg/L), median (IQR)	1.00 (0.48–2.26)	2.35 (1.17–4.25)	<0.001
ALT (U/L), median (IQR)	15 (12–20)	23 (17–32)	<0.001
AST (U/L), median (IQR)	21 (19–25)	25 (21–29)	<0.001
ALP (U/L), mean ± SD	63 ± 19	72 ± 22	<0.001
GGT (U/L), median (IQR)	19 (13–27)	39 (27–60)	<0.001
eGFR (mL/min/1.73 m^2^), median (IQR)	95.9 (84.2–105.7)	88.4 (76.4–99.4)	<0.001
UAE (mg/24 h), median (IQR)	7.4 (5.6–11.0)	9.7 (6.6–17.3)	<0.001
Total cholesterol (mmol/L), mean ± SD	5.31 ± 1.00	5.71 ± 1.03	<0.001
Non-HDL cholesterol (mmol/L), mean ± SD	3.96 ± 0.98	4.60 ± 1.00	<0.001
LDL cholesterol (mmol/L), mean ± SD	3.49 ± 0.90	3.79 ± 0.92	<0.001
HDL cholesterol (mmol/L), mean ± SD	1.35 ± 0.30	1.11 ± 0.24	<0.001
Triglycerides (mmol/L), median (IQR)	0.94 (0.71–1.26)	1.66 (1.26–2.19)	<0.001
Total BCAAs (µM), mean ± SD	356.90 ± 62.58	425.48 ± 67.78	<0.001
Valine (µM), mean ± SD	197.22 ± 33.20	229.74 ± 35.48	<0.001
Leucine (µM), mean ± SD	119.78 ± 23.64	144.48 ± 27.63	<0.001
Isoleucine (µM), mean ± SD	39.74 ± 13.47	51.19 ± 15.99	<0.001

Data are given in number with percentages (%), mean ± standard deviation (SD) for normally distributed data or median with interquartile ranges (IQR) for non-normally distributed data. Abbreviations: ALP, alkaline phosphatase; ALT, aminotransferase; AST, aspartate aminotransferase; BCAA, branched-chain amino acids; BMI, body mass index; FLI, fatty liver index; eGFR, estimated glomerular filtration rate; GGT, gamma-glutamyltransferase; HOMA, Homeostasis Model Assessment; HDL, high-density lipoproteins; hsCRP, high sensitivity C-reactive protein; IR, insulin resistance; LDL, low-density lipoproteins; MetS, metabolic syndrome; T2D, type 2 diabetes; UAE, urinary albumin excretion. LDL cholesterol was calculated by the Friedewald formula.

**Table 2 nutrients-11-00705-t002:** Multivariable linear regression analysis demonstrating the positive association of plasma branched-chain amino acids with an elevated fatty liver index (FLI) (≥60) after adjustment for clinical and laboratory covariates in 5791 subjects.

	Model 1		Model 2		Model 3		Model 4		Model 5	
	β	*p*	β	*p*	β	*p*	β	*p*	β	*p*
Age	0.015	0.173	0.014	0.199	0.010	0.339	−0.032	0.035	−0.046	0.002
Sex (men vs. women)	0.424	<0.001	0.426	<0.001	0.426	<0.001	0.431	<0.001	0.431	<0.001
FLI ≥60 vs. <60	0.326	<0.001	0.324	<0.001	0.323	<0.001	0.318	<0.001	0.208	<0.001
Family history of T2D (yes/no)			0.045	<0.001	0.048	<0.001	0.046	<0.001	0.033	0.002
Alcohol intake (≥10 g/day)					0.007	0.495	0.011	0.325	0.014	0.207
Current smoking (yes/no)					−0.043	<0.001	−0.044	<0.001	−0.030	0.005
eGFR (mL/min/1.73 m^2^)							−0.057	<0.001	−0.046	0.002
UAE (mg/24 h)							−0.015	0.165	−0.021	0.058
Use of antihypertensive medication							0.009	0.478	−0.011	0.376
Use of lipid-lowering drugs							0.026	0.026	0.010	0.395
HOMA-IR									0.271	<0.001
HOMA-β									−0.043	0.089

β: standardized regression coefficients. HOMA-IR and HOMA-β were log_e_ transformed for analyses. eGFR, estimated glomerular filtration rate; FLI, fatty liver index; HOMA, Homeostasis Model Assessment; IR, insulin resistance; T2D, type 2 diabetes; UAE; urinary albumin excretion. Model 1: adjusted for age and sex. Model 2: adjusted for age, sex, family history of type 2 diabetes. Model 3: adjusted for age, sex, family history of type 2 diabetes, alcohol intake and current smoking. Model 4: adjusted for age, sex, family history of type 2 diabetes, alcohol intake, current smoking, estimated glomerular filtration rate, urinary albumin excretion and use of antihypertensive medication and lipid-lowering drugs. Model 5: adjusted for age, sex, family history of type 2 diabetes, alcohol intake, current smoking, estimated glomerular filtration rate, urinary albumin excretion, use of antihypertensive medication and lipid-lowering drugs, HOMA-IR and HOMA-β.

**Table 3 nutrients-11-00705-t003:** Prospective associations of FLI with incident type 2 diabetes.

	**Non-Elevated FLI (<60)**	**Elevated FLI (≥60)**	
Participants, *n*	4120	1671	
Incident T2D, *n* (%)	82 (2.0)	194 (11.6)	
	**HR (95% CI)**		***p*** **-Value**
Crude Model	(ref)	6.78 (5.23–8.79)	<0.001
Model 1	(ref)	5.84 (4.46–7.66)	<0.001
Model 2	(ref)	5.72 (4.37–7.50)	<0.001
Model 3	(ref)	5.64 (4.31–7.39)	<0.001
Model 4	(ref)	5.09 (3.77–6.88)	<0.001
Model 5	(ref)	3.84 (2.76–5.36)	<0.001
Model 6	(ref)	3.46 (2.45–4.87)	<0.001

Data are presented as hazard ratio (HR) with 95% confidence interval (CI). FLI, fatty liver index; BCAA, branched-chain amino acids; T2D, type 2 diabetes. Model 1: adjusted for age and sex. Model 2: adjusted for age, sex and family history of type 2 diabetes. Model 3: adjusted for age, sex, family history of type 2 diabetes, alcohol intake and current smoking. Model 4: adjusted for age, sex, family history of type 2 diabetes, alcohol intake, current smoking, eGFR, UAE, antihypertensive medication and lipid-lowering drugs. Model 5: adjusted for age, sex, family history of type 2 diabetes, alcohol intake, current smoking, eGFR, UAE, antihypertensive medication, lipid-lowering drugs, HOMA-IR and HOMA-β. Model 6: adjusted for age, sex, family history of type 2 diabetes, alcohol intake, current smoking, eGFR, UAE, antihypertensive medication, lipid-lowering drugs, HOMA-IR, HOMA-β and BCAAs.

**Table 4 nutrients-11-00705-t004:** Prospective associations of plasma total branched chain amino acids with incident type 2 diabetes.

	BCAA Per 1 SD Increment	*p*-Value
Participants, *n*	5791	
Incident T2D, *n* (%)	276 (4.8)	
	HR (95% CI)	
Crude Model	1.68 (1.57–1.81)	<0.001
Model 1	1.65 (1.52–1.79)	<0.001
Model 2	1.63 (1.50–1.77)	<0.001
Model 3	1.64 (1.50–1.79)	<0.001
Model 4	1.64 (1.49–1.81)	<0.001
Model 5	1.35 (1.20–1.53)	<0.001
Model 6	1.19 (1.03–1.37)	0.01

Data are presented as hazard ratio (HR) with 95% confidence interval (CI). FLI, fatty liver index; BCAA, branched-chain amino acids; T2D, type 2 diabetes. Model 1: adjusted for age and sex. Model 2: adjusted for age, sex and family history of type 2 diabetes. Model 3: adjusted for age, sex, family history of type 2 diabetes, alcohol intake and current smoking. Model 4: adjusted for age, sex, family history of type 2 diabetes, alcohol intake, current smoking, eGFR, UAE, antihypertensive medication and lipid-lowering drugs. Model 5: adjusted for age, sex, family history of type 2 diabetes, alcohol intake, current smoking, eGFR, UAE, antihypertensive medication, lipid-lowering drugs, HOMA-IR and HOMA-β. Model 6: adjusted for age, sex, family history of type 2 diabetes, alcohol intake, current smoking, eGFR, UAE, antihypertensive medication, lipid-lowering drugs, HOMA-IR, HOMA-β and FLI elevated (yes/no).

**Table 5 nutrients-11-00705-t005:** Mediating effect of BCAAs on the association of FLI with incident type 2 diabetes.

	Coefficient (95% CI) *	Proportion Mediated
Indirect pathway (ab path)	B = 0.040 (95% CI 0.027–0.054)	19.6% **
Total effect (ab + c’ path)	B = 0.204 (95% CI 0.174–0.232)	

Analyses were performed according to Preacher and Hayes Procedure. B: unstandardized regression coefficient. Coefficients are adjusted for age and sex. * 95% CIs were bias-corrected confidence intervals after running 2000 bootstrap samples. ** The size of the significant mediated effect is calculated as the standardized indirect effect divided by the standardized total effect multiplied by 100.

**Table 6 nutrients-11-00705-t006:** Mediating effect of BCAAs on the association of HSI with incident type 2 diabetes.

	Coefficient (95% CI) *	Proportion Mediated
Indirect pathway (ab path)	B = 0.033 (95% CI 0.026–0.041)	22.6% **
Total effect (ab + c’ path)	B = 0.146 (95% CI 0.115–0.176)	

Analyses were performed according to Preacher and Hayes Procedure. B: unstandardized regression coefficient. Coefficients are adjusted for age and sex. * 95% CIs were bias-corrected confidence intervals after running 2000 bootstrap samples. ** The size of the significant mediated effect is calculated as the standardized indirect effect divided by the standardized total effect multiplied by 100.

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
