# Peer review of "Non-Alcoholic Fatty Liver Disease and Risk of Incident Type 2 Diabetes: Role of Circulating Branched-Chain Amino Acids"

_nutrients, 2019, doi:10.3390/nu11030705_

Round 1
Reviewer 1 Report
In the abstract, the background description of the pathogenic mechanisms is not clear. Some believe that T2D precedes NAFLD. Or is there data in the literature to suggest that this pathway of metabolic diseases is more complex with some having NAFLD as a risk factor for T2D?
Also in the abstract: A little more information about what FLI is. What does ‘indirect pathway’ mean? Rephrase conclusion along the lines of, ‘The association of NAFLD with risk of developing of T2D seems like it could be mediated by elevated BCAA’.
In the introduction: Paragraph 1: It is mentioned that insulin resistance is important for the pathogenesis of NAFLD and then that T2D also precedes NAFLD. Is the former preclinical T2D before onset of hyperglycemia? This is a complex line of thinking, perhaps rephrase more cautiously by saying before these statements that although these variable findings are present, some studies have reported that suggest that NAFLD may be a risk factor for onset of metabolic diseases like T2D. Overall, the complexity of what is in the literature regarding the timing of T2D and NAFLD needs to be presented and that this might dependent on the population.
Results: If a sentence begins with a number, it must be spelled.
In the first paragraph of the discussion, the findings of the cross-sectional and longitudinal parts of the study need to be clearly addressed. For instance, was BCAA found to be already elevated in the subjects with NAFLD with no clinical T2D at baseline and then further increased in those that developed T2D? Furthermore, be clear what is mean by ‘joint effect’; for instance, was each finding of baseline NAFLD and elevated BCAA separately associated with development of T2D and both factors combined more likely predicted T2D?
The first paragraph does not outline in a mechanistic fashion what will be discussed throughout the discussion. Overall, the discussion does not flow by way of what is proposed to happen mechanistically based on the present findings. For example, begin the second paragraph with what is thought to occur earliest in the pathogenic process; are BCAA increased before NAFLD, etc? Also, strengthening the transitions between paragraphs will assist in highlighting this flow.
The conclusions in the discussion does not seem to clearly reflect the study design presented in the introduction. Was the main goal of the authors to determine if elevated BCAA was a marker of NAFLD and then whether the combined presence of this factors exaggerate the risk for T2D? The timing of this process is not clear in the introduction. This manuscript does to present a clear proposed pathogenic process to this reviewer.
Author Response
Response to comments of the reviewers
Reviewer 1
· In the abstract, the background description of the pathogenic mechanisms is not clear. Some believe that T2D precedes NAFLD. Or is there data in the literature to suggest that this pathway of metabolic diseases is more complex with some having NAFLD as a risk factor for T2D?
Response. In the revised manuscript we have rewritten the background and rationale of the present study now stating the NAFLD may precede T2D development. Indeed there is literature to support this supposition as delineated in the introduction section. The first sentences of the abstract now read: Non-alcoholic fatty liver disease (NAFLD) is likely to be associated with elevated plasma branched-chain amino acids (BCAA) and may precede the development of type 2 diabetes (T2D). We hypothesized that BCAA may be involved in the pathogenesis of T2D attributable to NAFLD, and determined the extent to which plasma BCAA influence T2D development in NAFLD.
· Also in the abstract: A little more information about what FLI is. What does ‘indirect pathway’ mean? Rephrase conclusion along the lines of, ‘The association of NAFLD with risk of developing of T2D seems like it could be mediated by elevated BCAA’.
Response. We have described the FLI algorithm in more detail in the abstract. The indirect pathway is a statistical term derived from mediation analysis, that indicates the extent to which the association of A (in this case and elevated FLI) with outcome (in this case incident T2D) is attributable to an intermediate variable (in this case plasma BCAA).The final sentence of the abstract now reads: The association of NAFLD with T2D development seems partly mediated by elevated BCAA.
· In the introduction: Paragraph 1: It is mentioned that insulin resistance is important for the pathogenesis of NAFLD and then that T2D also precedes NAFLD. Is the former preclinical T2D before onset of hyperglycemia? This is a complex line of thinking, perhaps rephrase more cautiously by saying before these statements that although these variable findings are present, some studies have reported that suggest that NAFLD may be a risk factor for onset of metabolic diseases like T2D. Overall, the complexity of what is in the literature regarding the timing of T2D and NAFLD needs to be presented and that this might dependent on the population.
Response. We appreciate this point of the reviewer. We have now rephrased the sentences indicating a close association of NAFLD with insulin resistance. Indeed while on the one hand, MetS and T2D may predict NAFLD development, two meta-analyses have shown- the other way around- that NAFLD predict incident T2D. This has been stated more clearly in the introduction section.
· Results: If a sentence begins with a number, it must be spelled.
Response. The corresponding sentence has been rephrased.
· In the first paragraph of the discussion, the findings of the cross-sectional and longitudinal parts of the study need to be clearly addressed. For instance, was BCAA found to be already elevated in the subjects with NAFLD with no clinical T2D at baseline and then further increased in those that developed T2D? Furthermore, be clear what is mean by ‘joint effect’; for instance, was each finding of baseline NAFLD and elevated BCAA separately associated with development of T2D and both factors combined more likely predicted T2D?
Response. We have rephrased the introduction paragraph of the discussion section clearly stating which findings are from the cross-sectional analysis and which from the longitudinal analysis. Concerning the “joint” effect of and elevated FLI and increased BCAA: we have rephrased this statement now reading: …suggest that NAFLD and elevated circulating BCAA could act together on T2D development.
· The first paragraph does not outline in a mechanistic fashion what will be discussed throughout the discussion. Overall, the discussion does not flow by way of what is proposed to happen mechanistically based on the present findings. For example, begin the second paragraph with what is thought to occur earliest in the pathogenic process; are BCAA increased before NAFLD, etc? Also, strengthening the transitions between paragraphs will assist in highlighting this flow.
Response. In the revised discussion section we have started by discussing the cross-sectional association of plasma BCAA elevations with an elevated FLI and an elevated HSI. This is now more clearly indicated. The present results are discussed against the pathophysiology of what is known about the pathophysiology of hepatic fat accumulation, oxidative stress, altered BCAA metabolism and the effects thereof on oxidative stress and TCA metabolism. The following paragraph now puts our longitudinal data into perspective of earlier results from clinical studies. Further, as it seems likely that altered BCAA metabolism and hepatic fat accumulation act together in worsening of NAFLD/NASH on the one hand and further deteriorating BCAA metabolism on the other hand. For these reasons it is not quite possible to indicate which abnormality comes first.
· The conclusions in the discussion does not seem to clearly reflect the study design presented in the introduction. Was the main goal of the authors to determine if elevated BCAA was a marker of NAFLD and then whether the combined presence of this factors exaggerate the risk for T2D? The timing of this process is not clear in the introduction. This manuscript does to present a clear proposed pathogenic process to this reviewer.
Response: As it stands we feel that the changes made in the manuscript now leads the reader through a logic chain of thoughts. To accommodate the concern of the reviewer we have modified the conclusion paragraph at the end of the discussion section now reading: In conclusion, this large-scale population study cross-sectionally demonstrated that elevated plasma BCAA levels are positively associated with an elevated FLI, as a proxy of NAFLD. Furthermore, it was longitudinally shown that the association of NAFLD with T2D development is likely to be attributable in part by plasma BCAA elevations which likely reflect abnormalities in BCAA metabolism.

Reviewer 2 Report
In line with the Authors' study limitations in the Discussion section, this Reviewer considers that this type of study is not true original research article containing lots of bias.
Despite that, this manuscript sounds well-written and highly-motivated. At the very least, the Authors should
1. Create a diagram (i.e., schematic presentation) depicting the Authors’ critical insights and new findings from this study. It will be helpful to our readers.
Author Response
Response to comments of the reviewers
Reviewer 2
· In line with the Authors' study limitations in the Discussion section, this Reviewer considers that this type of study is not true original research article containing lots of bias.
Response: We present a large-scale cross-sectional and prospective analysis regarding the association of plasma BCAA with NAFLD, using the FLI and HIS as proxy of NAFLD. Second, we present a longitudinal analysis with respect to the association of and elevated FLI and plasma BCAA levels on incident type 2 diabetes. In our opinion, our study is solid, and the data presented are internally consistent. As with every epidemiological studies of this kind, the associations found are not per se causative.
· Despite that, this manuscript sounds well-written and highly-motivated. At the very least, the Authors should Create a diagram (i.e., schematic presentation) depicting the Authors’ critical insights and new findings from this study. It will be helpful to our readers.
Response: We appreciate the comments of the reviewer who states that our manuscript sounds well written. In the revised version of the manuscript we have now included a graphical abstract highlighting the main conclusions of our paper.
Graphical abstract concerning manuscript Non-Alcoholic Fatty Liver Disease and Risk of Incident Type 2 Diabetes: Role of Circulating Branched-Chain Amino Acids
Cross-sectional (n=5,791; no Diabetes at baseline) | Plasma Branched Chain Amino Acids |
Elevated Fatty Liver Index | Increased |
Elevated Hepatic Steatosis index | Increased |
Association with elevated Fatty Liver Index (fully adjusted) | Positive |
Association with elevated Hepatic Steatosis Index (fully adjusted) | Positive |
Longitudinal (n=5,791; follow-up 7.3 years) | Incident Type 2 Diabetes (n=276) |
Association with elevated Fatty Liver index (fully adjusted) | Positive; Hazard Ratio 3.46 |
Association with elevated Hepatic Steatosis index (fully adjusted) | Positive; Hazard Ratio 2.29 |
Association with Plasma Branched Chain Amino Acids | Positive |
Proportion of Branched Chain Amino Acids on effect of elevated Fatty Liver Index | 19.6 % |
Proportion of Branched Chain Amino Acids of effect of elevated Hepatic Steatosis Index | 22.6 % |
Conclusion 1: Plasma Branched Chain Amino Acids are associated with elevated Fatty Liver Index and Hepatic Steatosis Index as proxy of Non Alcoholic Fatty Liver Disease |
Conclusion 2: Elevated Fatty liver Index and Hepatic Steatosis Index predict incident type 2 diabetes mellitus. This association is in part attributable to increased Branched Chain Amino Acids |
